# Attachment Stories in Middle Childhood: Reliability and Validity of Clinical and Nonclinical Children’s Narratives in a Structured Setting

**DOI:** 10.3390/ijerph19159053

**Published:** 2022-07-25

**Authors:** Jolien Zevalkink, Elle Ankone

**Affiliations:** 1Department of Clinical, Neuro- & Developmental Psychology, Faculty of Behavioural and Movement Science, Vrije Universiteit, van der Boechorststraat 7, 1081 BT Amsterdam, The Netherlands; 2Department of Education & Pedagogy, Faculty of Social and Behavioural Sciences, Utrecht University, 3584 CH Utrecht, The Netherlands; e.ankone@engh.nl

**Keywords:** attachment narratives, middle childhood, clinical assessment

## Abstract

Middle childhood is one of the most understudied periods of development and lacks a gold standard for measuring attachment representations. We investigated the reliability and validity of a Dutch version of the Story-Stem Battery coded using the Little Piggy Narrative (LPN) Coding System in a clinical (*N* = 162) and a nonclinical group (*N* = 98) of 4–10-year-old children. Their attachment stories were furthermore coded using the coherence scale. Factor analyses showed that the items of the LPN system formed four attachment scales and a separate scale reflecting distress/anxiety, with sufficient internal consistency for the scales and high interrater reliability (*n* = 20). Furthermore, we studied construct and discriminatory validity. The attachment scores correlated with coherence and child behavioral problems in the expected direction. Results showed age and gender differences, indicating that separate norm groups are necessary. In particular, disorganized attachment, coherence and distress/anxiety differ between clinical and nonclinical children across age and gender. Results for the other three organized attachment scales were more complex. For instance, older boys from the nonclinical group had higher scores on secure attachment than their clinical peers, while girls from the clinical and nonclinical groups did not differ, even though girls in the nonclinical group had higher secure attachment scores than boys. Results are discussed in light of attachment theory and developmental pathways in middle childhood, as well as their clinical implications.

## 1. Introduction

Understanding the internal working model of children from an attachment perspective may be a useful transdiagnostic entry point for treatments that focus on improving the relational functioning of children with insecure attachment histories and their parents. In early childhood, the quality of attachment relationships is assessed by observing how children respond behaviorally to stressful situations. More specifically, researchers assess how young children use their parents as a secure base from which to explore and a secure haven for support when feeling insecure [1]. In middle childhood (age 4–10 years), children’s cognitive development enables them to gradually form attachment representations or internal working models of self and others [2,3,4,5,6,7,8,9,10,11]. More and more empirical research is accumulating to substantiate this hypothesis of a gradual move to the level of representation before a more robust internal working model of self and others is formed [12]. By studying the narratives of children about relevant social and emotional themes, for instance, in the form of autobiographical memories, it becomes possible to study these affective meaning-making processes [13,14,15,16,17]. In addition, this gradual process in middle childhood provides a window of opportunity to prevent insecure attachment patterns to develop into structural mental health problems at a later age. 

From an attachment perspective, middle childhood is a relatively understudied period and lacks a gold standard for measuring the quality of children’s social and emotional interpersonal relationships [18]. Because of this move to the level of representation during this period, an attachment instrument should focus both on observable attachment behavior and the internal working model [19]. The Attachment Story Completion Task (ASCT), first developed by Bretherton, Ridgeway, and Cassidy [13] for preschoolers, focuses both on verbal and nonverbal behavior in relation to attachment issues presented in the story stem. Since its development, the ASCT has been used in both clinical and nonclinical settings [20]. However, the ASCT is not considered as well researched as the Strange Situation or Adult Attachment Interview [18]. Other attachment instruments in middle childhood are more dependent upon the verbal skills of children, such as the Childhood Attachment Interview [21,22,23] and the Attachment Script Assessment, which was recently adapted for this age period [24]. In addition, self-report questionnaires, such as the Security Scale [25], depend on the ability of the children to reflect on their own behavior as well as the behavior of their caregivers, which is often compromised in clinical populations and not fully developed throughout middle childhood [18,19]. This paper aims to examine the usefulness of the ASCT as an assessment instrument for younger (age 4–7 years) and older (age 8–10 years) children in middle childhood by empirically identifying the four attachment patterns (secure, ambivalent, avoidant, disorganized) [1,26], using continuous scores for each attachment scale.

*Secure* attachment represents the ability to create coherent, qualitatively rich autobiographical memories [27,28,29]. The richness of self-knowledge and autobiographical memory is mediated by the way “caregivers co-construct narratives about external events and the internal, subjective experiences of the characters” [30] (p. 58). In child narrative assessments, children are presented with story beginnings or stems and asked to complete the story using standardized administration. The quality of the attachment stories of children in middle childhood is related to their self-organization, defined as “an innate property that creates a sense of order, cohesion, and stability over time” [30] (p. 195). A child will develop certain, sometimes conflicting, self-descriptions (e.g., friendly, curious, scared, stupid) based on experiences with others with whom the child has developed stable emotional relationships. “Children tend to portray the dolls in the narratives and the parent-child relationships in a way that corresponds to their self-image and their own relationships” [20] (p. 223). Securely attached children address the story issues openly and produce benign resolutions, depicting adults as caring and children as competent. One of the main advantages of a secure attachment pattern is that it provides a fertile ground for integrating different self-images by providing a sense of coherence [29,30,31]. Healthy self-organization makes it easier to reflect or communicate with others and co-construct narratives that form new, coherent autobiographical memories across the lifespan [29,32]. 

In insecure attachment, the process of self-organization may become inhibited when rigid or chaotic self-states are dominant [30] (p. 216). For *insecure-ambivalently* attached children, self-images are related to their need for support, exaggerating or enlarging their emotional state, and an implicit lack of self-confidence, exemplified by statements such as “wanting to stay close to my mom” and “I often get angry.” Uncertainty about caregivers’ reactions due to their experiences with sudden, insensitive, often disruptive ways of communicating has left implicit messages about self-images such as “not always worthy of attention” or “unable to self-soothe.” [30,33]. These negative models of the ambivalently attached children’s self-images reflect past experiences with inconsistent caregiver messages. Self-organization in children with an *insecure-avoidant* attachment history may be characterized by attentional inflexibility, idealization, and/or disconnection to others [30,34]. In the attachment narratives of children with externalizing problems, which are often associated with avoidant attachment, Warren [20] described themes of disobedience, danger, preoccupation with food, and having the doll figure acting like a superhero. Their self-images seemed focused on non-emotional domains, such as “good soccer player” or “very strong,” and autobiographical memories often lacked details and vitality, as if experienced from a distance [35].

In the case of *insecure-disorganized* attachment, mental models of self, or self-images include frightened, frightening or disorienting aspects often leading to feelings of dissociation, and disconnection. For instance, the self may be considered “scared,” “like a wild animal,” or “deserving to be punished.” Attachment stories of children with a history of abuse have been found to contain elements of physical or sexual abuse and negative self-images of dolls representing children and adults, whereby these dolls often experience accidents, get hurt, or die [20]. Incongruent, incoherent, or fragmented mirroring by caregivers has forced the child to internalize the mental states of the caregiver as part of their self-identity, even though these are not contingent with their own self-images [5,34,36,37,38]. Nevertheless, these processes can be changed by entering circumstances that enhance more positive models of self and others. Adoption research has shown that attachment representations in the story stems of formerly maltreated children can change for the better, even for late-adopted children, thanks to adoptive parents classified as secure-autonomous [39,40,41,42].

The ASCT originally focused on *story stem battery’s* pertaining to attachment themes between parent and child [13]. Later, story stem assessments also included other social and emotional themes, such as moral dilemmas, aggressive themes, or the handling of divorce [43,44,45]. Furthermore, different batteries have been developed, including the MacArthur Story Stem Battery (MSSB) [44] and the Manchester Child Attachment Story Task [46], adapted for different goals or populations [47]. Story stem *coding systems* have also varied widely. A review by Page [45] into the use of story stem assessments showed that only three of the 11 research projects coded attachment or computed an attachment score. The majority of projects focused on ratings of content themes, such as positive coping with separation from parents versus avoidance. For instance, Belden and colleagues [48] found that preschoolers’ negative and disciplinarian representations of their mothers during the story stems were related to their mothers’ non-supportive behaviors and negative affect one year later.

When story stem coding systems did include attachment, its operationalization also varied considerably. Some have used a single attachment-security rating scale with cut-off scores for insecurity. Others included a rating of insecurity and disorganization, alongside content ratings [49]. Most coding systems have used a categorical system with children falling in one of the attachment categories. A few coding systems have used four continuous attachment scores for each child, such as the attachment Q-sort for narratives [50,51], or dimensional scales with different categorical profiles [52]. The LPN-story stem battery and coding system have been developed on the basis of clinical experience to assess changes in the attachment representations of children with maltreatment history after adoptive placement [39,42]. Hodges and colleagues combined coding content items, such as “adult provides help,” for each story stem with attachment scores across stems. The scores on the four theoretically-based attachment constructs for each child were calculated based on the coding of the content items for each story stem. Both the use of content items and the training to become reliable on coding content was considered relevant for clinical practice. This may also be called bottom-up coding compared to top-down coding of attachment classifications. This approach also resulted in four continuous attachment scores. Having these scores available at the beginning of treatment and periodically throughout, allows the mental health professional to assess changes in scores from more secure to less disorganized attachment patterns. Their attachment constructs have not been empirically validated yet. The present study examined whether empirical and meaningful attachment scales could be constructed from these clinically relevant content items. To begin, we first give a short overview of narrative studies in middle childhood that have used attachment scores, a coherence rating scale, and assessments of problem behavior in nonclinical and clinical groups.

### 1.1. Nonclinical or Typically Developing Participants

Two studies used a five-point rating scale for attachment security to place children in one of four attachment classifications. The first study found significant and positive relations between the child’s working model of the self (Puppet Interview) and attachment to mother [53]. The results of the second study showed that children with avoidant attachment representations had more conflictual and less close relationships with their teachers in preschool, as well as less prosocial orientation towards their peers than children with secure attachment representations [54]. A longitudinal study of attachment development into middle childhood included attachment measures at three points in time, including the ASCT [2]. The authors reported a strong concordance (74%) between the quality of secure attachment relations in infancy and attachment security assessed with the ASCT (secure, avoidant, hostile/negative) at five years of age. A study into attachment security and adjustment to school with 10-year-old children investigated both the Bretherton attachment classification (secure, fairly secure, insecure) and the coding according to four attachment prototypes on a five-point rating scale across the stems [55]. Their results showed a normative distribution of the children based on the prototypes, with some significant gender differences: girls were more often classified into secure and ambivalent attachment categories and less often into avoidant and disorganized attachment compared to boys.

Coherence in attachment narratives has been studied and related to child behavior problems in nonclinical groups. In narratives, coherence is “the ability to report attachment-related experiences in a clear, logical, affectively regulated manner” [56] (p. 710). Studies have shown narrative coherence (ASCT) to be negatively correlated to externalizing child behavior problems according to their mothers [17], in particular for girls [57]. However, Laible and others [58] found that coherence was only negatively related to the level of externalizing symptoms according to teachers of six-year-olds, but not related to their mothers’ CBCL reports. For preschoolers with more incoherent narratives (MSSB), a stronger relation was found between maternal stress and their mothers’ reported level of internalizing symptoms in contrast to children who told more coherent narratives [59]. In a study by Moss et al. [60], attachment was measured with a separation-reunion task at age 5–7 years and coherence two years later as part of the MSSB. Their results showed that children in the secure group had higher coherence scores than children in the ambivalent group and that girls had higher coherence scores than boys. They also reported that the 10-point Coherence Scale had an interrater reliability of 0.84 (ICC).

### 1.2. At-Risk or Clinical Participants

In a group of African-American women at risk of depression (40% above clinical cut-off) and living in at-risk circumstances (teenage pregnancy, low-income household), maternal sensitivity significantly predicted their preschoolers’ level of attachment security [61]. The majority of these children (52%) had been rated as anxious or disorganized on the ASCT using a 10-point rating scale. Poehlmann [62] showed that 68% of children (age 2–7 years) whose mothers were incarcerated had insecure attachment representations. In addition, a significant difference was found between children living in institutions versus those living with their families in attachment security (8-point rating scale), with lower security of attachment and a higher proportion of disorganized narratives for institutionalized children [63]. The Dutch ASCT-LPN was used to assess attachment representations in children with a mild intellectual disability (IQ > 50 < 85) and results showed it to be a reliable instrument [64]. A meta-analysis combined clinical and nonclinical populations from middle childhood, whereby the researchers found small to moderate associations between avoidant attachment and internalizing behavior, with 59% of the studies using the CBCL or TRF to assess behavior problems [65]. Other studies have shown a relation between insecure attachment, in particular avoidant and disorganized, and externalizing behavior problems in middle childhood [56,63,66,67].

In the present study we investigated the psychometric qualities of the attachment story stem battery and the Little Piggy Narrative (LPN) Coding System [39,68] in a clinical and a nonclinical group of children in middle childhood. First, we investigated the underlying constructs of the coding system in a clinical population by assigning the content items to attachment scales that resulted from carrying out a Principal Component analysis and by relating these empirically derived scales to theory-informed LPN-attachment constructs. We expected to find meaningful empirical attachment scales [53,61,63]. Second, we investigated the internal consistency of the constructed scales and the interrater reliability of the LPN coding system. In line with previous studies, we expected coding to be trainable and transferable between coders. Third, we studied the discriminatory and construct validity of the constructed scales. In order to study discriminatory validity, we examined differences between age groups, gender (boys/girls), and setting (clinical/nonclinical) for the constructed scales and coherence. We expected that older children would construct more secure and coherent narratives compared to younger children [60]. Additionally, we expected gender differences for coherence [21,57,60] and the continuous attachment scores [51]. We also expected differences between children from the clinic and the community in attachment and coherence ratings based on general findings of higher prevalence of disorganized attachment in clinical populations [17,56,69]. For construct validity, we examined the interrelations between the four attachment scales for children in the two settings, the distribution of children into attachment categories, and the relation with the coherence ratings and the level of child behavior problems according to the mothers. We expected secure attachment to relate negatively to the three insecure scales. In addition, we expected the distribution of children into the four attachment categories in middle childhood to be comparable to that found in other clinical and nonclinical groups [55,62,63,69]. For coherence, we expected it to be positively related to secure attachment and negatively to insecure attachment [60]. With regard to children’s level of problem behavior, we expected that mothers of children with insecure attachment would report more externalizing problems [16,33,70], but inconsistent results have also been reported [49,65]. These findings will be discussed in light of implications for attachment theory, developmental pathways in middle childhood, and the usefulness of the ASCT as an assessment instrument in clinical practice.

## 2. Materials and Methods

### 2.1. Participants and Procedures

Two groups of children (age 4–10 years) comprised the study participants. One group was obtained from a clinic (*N* = 162; 61% boys) and the other, which we will refer to as the nonclinical group, recruited from the community (*N* = 101; 44% boys). For the clinical group, no attrition rates were available. In the nonclinical group, ASCT data were missing for four children (2 children ill, 1 child refused to participate, 1 unknown), which resulted in a group of 98 children. Children in the clinical group had been referred to a diagnostic assessment for psychodynamic treatment due to their complex social and emotional problems. Children in the nonclinical group were recruited via local primary schools to form a control group for another study by the second author examining the relation between problems with math computation bordering dyscalculia, quality of attachment, and stress responses. In the clinical group, the average age was 7.9 years (*SD* = 1.74), with boys being significantly younger than girls, *t* (160) = 2.04, *p* = 0.043 (resp.: *m* = 7.6, *SD* = 1.65; *m* = 8.2, *SD* = 1.83). The average age in the nonclinical group was 9.1 years of age (*SD* = 1.13) with no significant age differences between boys and girls. Because of the broad age range, two age groups were formed: younger children (4–7 yrs.) and older children (8–10 yrs.).

Informed consent was obtained from the parents of all participants. In the clinic, data were gathered during routine assessments in which parents were asked to share information anonymously for scientific purposes via an informed consent. For the nonclinical group, the second author obtained permission for the data collection from the Ethical Review Board at Utrecht University (March 2014). 

*Clinical Group*. The ASCT was part of a regular assessment when children (and their parents) were applying for psychodynamic treatment in an outpatient setting. The full assessment took place over two mornings and included other instruments, such as the CBCL and WISC. The intake team also gave DSM diagnoses to the children. For the clinical group, DSM-IV codes were available, which have been arranged in line with the DSM-5 diagnostic categories (e.g., separation anxiety disorder (309.21) from Childhood disorders to Anxiety disorders) [71]. Descriptive analysis showed that the largest group of children (*n* = 45; 27.2%) was diagnosed with parent–child relational problems (V61.20), which falls in DSM-5 under “Other conditions that may be a focus of clinical attention (715)”. The distribution into the main DSM-5 diagnostic categories is shown in Figure 1. The right bar includes 11 children (6.8%) that fell in the category “unspecified axis I,” which is no longer used in DSM-5. 

*Nonclinical Group*. An invitation to participate was distributed at five local primary schools with which the second author was acquainted due to her role as a registered psychologist with a private practice. Participation was voluntary and without compensation. The children participated in a three-hour assessment conducted by trained research assistants from Utrecht University. The assessment included physiological measurements during calculating tasks, the attachment story stems, and a social stress task. A few weeks after the assessment, parents received a summary report about their child’s calculating level, stress level, and attachment style. 

### 2.2. Measures

#### The Attachment Story Stem Completion Task (ASCT) 

*Story Stems Battery*. The ASCT aims to assess children’s perceptions of the quality of relationships between parents and children by providing them with story stems that trigger their attachment needs for protection, comfort, and parental awareness, such as feeling hurt, lost, or excluded [13,53]. After each emotionally evoking story stem, children are asked with a neutral voice “to show me and tell me what happens next?” This allows children to use both verbal and nonverbal means of communication. An example of a story stem is Burnt hand: a mother urges her impatient child to wait while she prepares dinner with the eventual consequence that the child reaches for the stove and ends up with a burnt hand, while the younger sibling and father sit at the table [13,39]. The Dutch ASCT is composed of nine stems, of which five stems have been developed for a clinical population by Hodges et al. [39]: Crying outside, Little pig, Stamping elephant, Picture from school, and Bikes. In addition to these five stems, four stems from the MSSB [13] were selected that differentiated between adopted, maltreated children, and nonclinical children [39]. These four stems are: Burnt hand (also called Hot soup), Lost keys, Burglar in the dark (also called Monster in the dark), and Exclusion (for description and modifications, see [39]). In the nonclinical group, the following five stems were used as part of the larger assessment: Crying outside, Stamping elephant, Picture from school, Burnt hand, and Burglar in the dark. One story stem only contains animals (Little pig), while the other story stems include a standard doll family (mother, father, older child, younger child). The family does not replicate children’s own family configuration to allow them to represent their experiences in displaced form [39]. The stories were recorded on video for coding at a later time. Administration of a set of nine story stems takes about 35 min and a set of five stems about 20 min.

*Coding System*. The children’s responses to the story stems were coded using the Little Piggy Narrative (LPN) Story Stem Coding Manual [68], which contains 37 items that are coded for the children’s responses to each story stem separately. Each item is scored with a three-point rating scale (0 = absent, 1 = sometimes/somewhat present, 2 = present). Of the 37 items in the LPN coding system, the first is: No engagement. If a child does not respond to the story stem, they will receive a score of 2 and the remaining 36 items are not coded. Story stems with responses on the 36 items were included in further statistical analyses and children had to complete a minimum of four stems. The development of the LPN system has gone through several stages. When the theoretical distribution of the LPN-items into four attachment constructs occurred, 32 items were used. At a later date, five items were added to their original coding system [39]: no closure, adult shows aggression, child shows aggression, physical punishment, and repetition. To compare the empirically constructed attachment scales and the theoretical attachment constructs (LPN-security, LPN-insecurity, LPN-defensive/avoidance, LPN-disorganized), we could only use the 32 items, but for all remaining results, all 36 LPN-items were used. The first author had passed the accreditation standards on the 3-point LPN coding system with an overall interrater reliability of 82% and trained the assessors at the clinic. After becoming a certified coder, coding and calculating the responses of children to the ASCT takes about 45–60 min per administration.

### 2.3. Coherence

The Coherence Scale aims to measure the level in which children openly discuss both positive and negative emotional themes related to the attachment story stems and organize these themes in a coherent, well-resolved narrative, showing that they understand the conflict and offer a resolution that includes embellishment without incoherent segments. Coherence was coded for each story stem on a 9-point scale in which we combined the narrative coherence and emotional coherence scales developed by Oppenheim and colleagues [17] (p. 287). The scale ranged from 1 (fragmented, shifts in story line without explanations, bizarre or chaotic segments/emotions/cognitions) to 9 (child understands the conflict and offers a resolution that includes embellishment; there are no incoherent segments/emotions/cognitions). The average coherence score was calculated across all stems for each of the children. Oppenheim and colleagues reported an internal consistency of 0.87 at age 4 and 0.80 at age 5 for the narrative composite, which was generated by summing the scores across narratives. We examined interrater reliability for the nonclinical group (see results).

### 2.4. Behavioral Problems

The level of children’s behavioral problems was assessed by mothers’ responses to the Dutch Child Behavior Checklist (CBCL), which is a widely used instrument with sufficient psychometric quality [72]. The instrument consists of three main scales (Internalizing, Externalizing, Total Problems), eight syndrome scales (e.g., anxious/depressed, somatic complaints), and six DSM-oriented scales (e.g., attention deficit/hyperactivity problems, conduct problems). Table 1 shows that children in the clinical group had higher average T-scores for the internalizing, externalizing, and total problems scales than the nonclinical group.

### 2.5. Statistical Analysis

We first conducted several principal component analyses (PCA) with varimax (Kaiser normalization) rotation entering all 36 items for each of the nine story stems to investigate whether empirical attachment scales could be found. The PCA’s were conducted with the clinical group, which contained a sufficient number of children (>150). Furthermore, intercorrelations between items were significant and often larger than 0.30. In addition, Bartlett’s test of sphericity was significant (*p* = 0.00), the Kaiser–Mayer–Olkin value was 0.77, and the assumption of multicollinearity was not violated. Next, we examined the reliability of the constructed scales and Coherence Scale using Cronbach’s alpha for internal consistency and Pearson correlations for the interrater reliability (IRR). To examine the differences between age and gender groups, we used the independent-samples *t*-test procedure. For the differences between clinical and nonclinical groups, we used one-sample *t*-tests and entered the corresponding average from the community group in the test value box. We examined the intercorrelations and correlation with coherence using the Spearman’s rho correlation coefficient, because we expected the distribution of the scales to be skewed. Next, to examine the distribution of children into the four attachment categories, we divided the continuous scores on each attachment scales into three equal groups (low, middle, high) using cut-off points. The proportion of children was calculated based on a ranking procedure explained under results. To examine the correlations between the attachment scales (secure, ambivalent, avoidant, disorganized) and CBCL main scales (Internalizing, Externalizing, Total problem) and subscales, we used Spearman’s rho correlations. Finally, we examined differences on the CBCL between children with high and low scores on the attachment scales by categorizing the children for each scale into two groups using a median split and performing the *t*-test procedure for independent samples to approach categorical coding of attachment in line with previous studies.

We conducted all analyses using SPSS version 27. A *p*-value of <0.05 was used to indicate statistical significance.

## 3. Results

### 3.1. Reliability: Data Reduction of the LPN-Items

#### Factor Analysis

In order to construct empirically derived attachment scales from the LPN-items, we conducted several principal component analyses with varimax (Kaiser normalization) rotation. In the first step, 36 items were included and four factors were extracted. This did not clearly produce the expected four-factor solution. The scree plot rather suggested three factors. The results also showed that the items representing disorganization in the LPN-system were distributed across factors with both secure and insecure LPN-items. We excluded these six items from further factor analyses, but did use them to form a separate “disorganized” scale. This is in line with other attachment assessments, where a score on a nine-point rating scale of disorganized attachment is obtained in addition to a categorical assignment to on one of the three attachment classifications (secure, ambivalent, avoidant). Three items with positive loadings on both secure and insecure factors were excluded from the factor analyses, but were used to form a separate scale representing a high level of distress and anxiety. Finally, three items had low factor loadings (<0.40); these items were excluded from further analyses. A principal component analysis with the remaining 24 items provided a three-factor solution explaining 43.9% of the variance, with 22.4% for the first factor, 14.6% for the second factor, and 6.9% for the third factor. The LPN-items and factor loadings are shown in Table 2, in which the items follow the order of the coding system and are not arranged according to the strength of the primary loading. Four items had loadings on two factors; three on both insecure-ambivalent and insecure-avoidant and one item on secure (reversed) and insecure-avoidant. As will be explained later, this overlap is in line with attachment theory.

The first factor explains 22.4% of the variance, consists of 11 items, and is labelled “insecure-ambivalent.” Six of these items describe threats to the attachment system in the form of danger, aggressiveness, and rejection for both children and adults (e.g., child endangered), three items relate to adults as the source of distress (e.g., adult actively rejects), and two items are about feeling stranded (no closure, repetition). Three of the items have loadings on both the ambivalent and avoidant scale (e.g., coherent aggression), which shows that the difficulty insecurely attached children have with conflict resolution and impulse control is still in the more coherent/organized range. The second factor explains 14.6% of the variance, consists of 11 items, and is labelled “secure.” It includes all six positive items (e.g., adult provides help/protection) and five items with negative loadings (e.g., disengagement) of which one item also loaded on another scale (adult unaware). The third factor explains 6.9% of the variance, consists of six items, and is labelled “insecure-avoidant.” These items relate to implicit and explicit avoiding/reducing of attachment stress from the narratives as the main strategy (e.g., denial/distortion of affect) by also acting aggressively as a child (e.g., child shows aggression).

### 3.2. Scale Construction

Based on the results of the factor analyses, scales were formed by adding the scores on the LPN-items that formed the factor (or reversed score in case of a negative loading) and dividing these by the number of items on the scale and the number of stories that the child had finished in order to make it comparable across children, irrespective of the number of stories within the assessment. Table 2 shows the LPN-items that form the first three scales. The fourth scale “disorganized” (six items) is similar to the theoretical LPN-construct. It contains the following items: Child “parents” or “controls” adult; Extreme aggression; Catastrophic fantasy; Bizarre/atypical responses; Bad-good shift; and Magic/omnipotence. The fifth scale, “distress/anxiety,” with three items, was also computed separately because these items loaded to both secure and insecure factors. It contains: Acknowledgement of distress or anxiety (CHILD); Acknowledgement of distress or anxiety (PARENT); and Neutralization/diversion from anxiety. For exploratory reasons, we added this scale alongside the statistical analyses with the attachment scales. Each child received a continuous score for each of these five scales.

#### Comparing the Three Empirical Attachment Scales with Theoretical LPN-Attachment Constructs 

The empirically derived attachment scales resemble the theoretical constructs underlying the LPN-coding system in several ways. The LPN-security construct contains 11 LPN-items. Seven of the 11 LPN-items were empirically supported in the secure scale, two items were used to form a separate stress/anxiety scale, one item (coherent aggression) from the LPN-security construct loaded on the LPN-insecurity construct, and one item (realistic active mastery) did not have a loading >0.40 and was excluded from the study. The LPN-insecurity construct consists of seven items. Five of the seven items were empirically supported by the ambivalent scale, one item (adult unaware) loaded higher on other scales, and one item (excessive compliance) did not have a loading > 0.40 and was excluded. The LPN-defensive/avoidance construct contains eight items. Two of the eight items were empirically supported by the avoidant scale, one item (no engagement) was excluded, three items (disengagement, initial aversion, premature foreclosure) had higher negative loading on the secure scale, one item (neutralization) correlated with stress/anxiety, and one item (avoidance within narrative frame) did not have a loading >0.40 and was excluded. In sum, some evidence was found for similarities between both data reduction methods, but the least overlap was found for the defensive/avoidance scale.

### 3.3. Reliability: Internal Consistency and Interrater Reliability

#### 3.3.1. Internal Consistency

Internal consistency was calculated for each of the scales with the LPN-items. Table 3 shows that most of the scales had sufficient reliability (≥0.70–<0.80) and the scale ambivalent had good reliability (≥0.80).

#### 3.3.2. Interrater Reliability

Two independent coders, of which one was the second author, coded 22 transcripts from the nonclinical group to establish interrater reliability (IRR) using Pearson correlations of the attachment scales and distress/anxiety. The results showed sufficient interrater reliability, ranging from 0.82 for secure attachment to 0.99 for disorganized. For coherence, data were available of 14 children and IRR was 0.88.

### 3.4. Discriminatory Validity: Age-Groups, Gender, and Setting

#### 3.4.1. Age-Groups

Results of *t*-tests for independent groups showed significant differences between younger and older children for secure, avoidant, coherence, and distress/anxiety scales, with older children having higher secure scores, lower avoidant scores, higher coherence scores, and higher distress/anxiety scores compared to younger children (Table 4). 

#### 3.4.2. Gender

Results of *t*-tests for independent samples showed significant differences between boys and girls for all scales, with boys having lower secure, coherence, and distress/anxiety scores and higher ambivalent, avoidant, and disorganized scores compared to girls (Table 5).

#### 3.4.3. Setting

As stated in the Section 2, we only compared older children (age 8–10 years) due to the significant age differences in the clinical group and the skewed distribution between clinical and nonclinical children across age. Before testing the differences between clinical and nonclinical groups, we examined differences between boys and girls within the nonclinical group (results for the clinical group in Table 5). Table 6 shows two significant differences between boys and girls from the nonclinical group for secure attachment and coherence, with higher scores for girls compared to boys.

One-sample *t*-tests were used to examine the differences between boys and girls from the clinical (resp., *n* = 42, *n* = 36) and nonclinical group. For boys, significant differences between clinic and community were found for secure attachment (*t* = −3.45, *p* = 0.001, *d* = −0.54), disorganized attachment (*t* = 3.23, *p* = 0.002, *d* = 0.50), coherence (*t* = −3.11, *p* = 0.003, *d* = −0.49), and distress/anxiety (*t* = −2.43, *p* = 0.020, *d* = −0.37). For girls, significant differences were only found for disorganized attachment (*t* = 5.55, *p* = <0.001, *d* = 0.93). The differences were in the expected direction, with higher scores for secure attachment and coherence, and lower scores for disorganized attachment, in the nonclinical group compared to the clinical group.

### 3.5. Construct Validity: Intercorrelations, Coherence, Categories, and Problem Behavior

#### 3.5.1. Intercorrelations between Attachment Scales 

Table 7 shows the *Spearman’s rho* correlations (due to skewed distributions) between the four attachment scales for the clinical group below the diagonal and the nonclinical group above the diagonal. In both groups, significant intercorrelations were found between the insecure scales, with relatively strong correlations between the disorganized and ambivalent scales. In addition, secure attachment was negatively related to avoidant attachment in both groups, showing that they represent a related but contrasting strategy. Tentatively, we also examined the intercorrelation between the attachment scales and the distress/anxiety scale in both groups. The distress/anxiety scale was significantly correlated to all attachment scores in the clinical group, but not in the nonclinical group. In the nonclinical group, a high score on distress/anxiety was only related to the secure and ambivalent scale. 

#### 3.5.2. Correlations between the four Attachment Scales and Coherence

*Spearman’s rho* correlations for both groups between the four attachment scales on the one hand and coherence on the other hand showed that coherence was positively correlated to secure attachment and negatively to all three insecure attachment scales in both the clinical and nonclinical group (Table 8). 

#### 3.5.3. Distribution of Continuous Attachment Scale Scores into Attachment Categories for the Clinical and Nonclinical Groups

To examine how the continuous scores of the four attachment scales related to the four attachment categories, percentile values were calculated by dividing each scale score into three equal groups (low, middle, high). To check for the distribution of children over the four attachment classifications, it was also necessary to rank the results for each scale. For instance, children who fell both in a high secure and high disorganized category were assigned to the disorganized classification in line with other attachment coding systems. For this ranking, we used the following order: disorganized, insecure organized (ambivalent + avoidant, ambivalent, avoidant), and secure. If children did not have a high score on one of the four attachment scales, they were placed in the category “no high score”. In the clinical group (*N* = 162), the distribution was as follows: 50 (30.9%) children had a high disorganized score, 9 (5.6%) children had high ambivalent/avoidant scores, 7 (4.3%) children had high ambivalent scores, 20 (12.3%) children had high avoidant scores, 27 (16.7%) children had high secure attachment scores, and 49 (30.2%) children had no high score on any attachment scale. To narrow down the number of children with no high score, we also included children with a medium score on the secure attachment scale, which resulted in 50 (30.9%) children assigned to the secure classification and 26 (16.0%) children not assigned to one of the four attachment categories. The distribution in the clinical group seems to be in line with other ambulant mental health settings, with more children in the insecure attachment categories compared to nonclinical populations and less compared to high-risk settings.

In the nonclinical group, the distribution was as follows: 11 (10.9%) children with high disorganized scores, 10 (9.9%) children with high ambivalent/avoidant scores, no children with high ambivalent scores, 10 (9.9%) children with high avoidant scores, 26 (25.7%) children with high secure attachment scores, and 41 (41.8%) with no high score. When we also included children with medium secure attachment, 52 (51.4%) children fell into the secure classification and 15 (15.3%) children were not in one of the four categories. We did not find a high proportion of children with only a high score on secure attachment, but we did find a normative proportion when including the medium score on the secure scale.

#### 3.5.4. Correlations between the Four Attachment Scales and Mothers Assessment of Children’s Behavioral Problems

First, *Spearman’s rho* correlations were calculated between the attachment scales and the three main CBCL scales (T-scores for internalizing, externalizing, and the total problem score) in both groups. In the clinical group, two significant correlations were found. The disorganized scale was negatively correlated to the CBCL-internalizing T-score and the Total Problems T-score (resp. r = −0.23, *p* = 0.005, CI [−0.38–−0.07]; r = −0.18, *p* = 0.033, CI −0.33–0.01). Unexpectedly, the results showed that children with higher disorganized scores had mothers who reported fewer internalizing problems for their children. In the nonclinical group, no significant correlations between the continuous scores on the attachment scales and the three main CBCL scales were found.

Next, the attachment scales were divided in high-low by median split and differences were examined for the three main CBCL scales, as well as for the syndrome and DSM-oriented subscales. The results showed that children with scores above the median of the secure scale had significantly less pervasive developmental problems, *t* (23) = 2.43, *p* = 0.023, according to their mothers. Children above the median of the ambivalent scale had significantly more problems related to the emotionally reactive syndrome subscale, *t* (23) = −2.09, *p* = 0.048. For children with high scores on the avoidant scale, no significant differences were found. For children with high scores on the disorganized scale, mothers reported lower scores on the syndrome subscales anxious/depressed, *t* (144) = 2.59, *p* = 0.011; somatic complaints, *t* (144) = 2.52, *p* = 0.013; and the DSM-oriented subscale affective problems, *t* (144) = 2.82, *p* = 0.005. For the syndrome CBCL subscales, such as emotionally reactive, only a relatively small number of scores (*n* = 24) were available. No significant differences were found for the main CBCL scales, except for children with high scores on disorganized attachment with mothers reporting lower internalizing problems, *t* (144) = 2.79, *p* = 0.006.

In the nonclinical group, children above the median of the secure scale had mothers who reported less problems on the main CBCL total problems scale, *t* (94) = 2.68, *p* = 0.009, and the internalizing scale, *t* (94) = 2.12, *p* = 0.037. For the CBCL subscales, significant differences in the same direction were found for the syndrome subscale anxious/depressed, *t* (94) = 2.65, *p* = 0.009, thought problems, *t* (94) = 2.54, *p* = 0.013, and the DSM-oriented subscale anxiety, *t* (94) = 2.69, *p* = 0.009, with mothers reporting less problems for children with secure ratings above the median. For children with a score above the median of the ambivalent attachment scale, a significant difference was found for the DSM-oriented subscale anxiety, with mothers scoring lower, *t* (94) = 2.00, *p* = 0.049. For children with an avoidant score above the median, mothers scored significantly higher on the DSM-oriented subscale conduct problems, *t* (94) = −2.32, *p* = 0.023. For children with a score above the median on disorganized attachment, mothers reported more withdrawn problems, *t* (94) = −2.17, *p* = 0.032.

## 4. Discussion

Overall, our study has found evidence for the psychometric quality of the attachment scales based on the Attachment Story Completion Task (ASCT) coded with the Little Piggy Narrative (LPN) Coding System. Three attachment scales (secure, ambivalent, avoidant) were empirically constructed and one scale (disorganized) needed to be constructed separately because these items loaded to both secure and insecure factors in line with other attachment assessments [26,73]. The attachment scales proved to have sufficient internal consistency and the interrater reliability was good. In addition, the attachment scales discriminated between younger and older children in middle childhood, boys and girls, and between clinical and nonclinical groups. The attachment scales correlated with coherence as expected. Relations with their mothers’ assessments of problem behavior were more in line with previous findings of attachment for children in the nonclinical than in the clinical group. Finally, we constructed an extra scale called distress/anxiety as it appeared relevant in the clinical sample. Next, we discuss these findings in more detail, in search of meaningful patterns and new insights to suggest possible improvements for the four attachment categories.

Data reduction of the LPN-items using principal component analysis yielded the following three attachment scales: secure, ambivalent, and avoidant. A high score on the secure scale shows narratives where children perceive adults and peers as a source of security, protection, comfort, and affection, and they feel at ease to ask for help or protection (e.g., child seeks help) with adults not being unaware of their problems. In addition, items that theoretically were assigned to avoidant attachment scored negatively on this scale, such as LPN-item 5 “premature foreclosure” (ending the story without resolution). The secure children approached the attachment conflict in the story stems with the intention of solving it instead of avoiding it. With regard to validity, our findings showed that secure attachment was related to higher coherence, in line with previous studies [60]. In addition, in the clinical setting, interesting age differences were found with older children having higher secure scores compared to younger children. It appears that older children more often have a secure script at hand as a strategic process [19]. When first using the ASCT in the clinic, we included children until 12 years of (mental) age, but soon restricted the age range to those who were ten years and younger, because otherwise many children showed secure attachment during the story stems. For older children, the Childhood Attachment Interview method seems a more reliable and valid method [18,21,23,74] to tap into their attachment representations. However, this issue may be less relevant for boys, because across middle childhood, they had lower scores on the secure attachment scale and the coherence rating than girls in both the clinical and nonclinical group. Additionally, other research has shown that girls express more secure representations and have higher coherence ratings than boys [51,60]. Nevertheless, for secure attachment in girls, no differences were found between settings, which means that this scale may have less clinical relevance for them as a sole indicator of psychological problems. Finally, the results of the forced distribution into four attachment classifications showed that 31% of the children in the clinic have a medium or high score on the secure scale compared to 51% in the nonclinical group, which is broadly in line with distributions of attachment classifications from other studies [74,75,76].

The ambivalent scale proved to be robust and explained a lot of the variance in the children’s narratives at intake in the clinical outpatient setting. A high score on the ambivalent scale shows narratives in which children perceive the world as a scary place for both children and adults, but also that adults increase these feelings of insecurity and are unaware of their problems. The children with high score on this scale are open about the aggression of the adults and their negative expectations concerning the safety of the world in an insecure but coherent manner. This seems to suggest that they are open to receiving treatment [77]. However, when distributing children based on high scores for this scale into attachment classifications, only a small proportion had a high score on this scale solely. Still, evidence for the validity of the ambivalent scale was also found in relation to coherence and distress/anxiety [33]. In particular, children with higher scores on the ambivalent scale showed more distress/anxiety in both the clinical and nonclinical group. In addition, their mothers reported more problems regarding emotional reactivity. However, this finding is tentative due to the small number of cases on which it is based.

The scale for avoidant attachment appears less reliable, because it only included six items with of a lot of overlap with the ambivalent scale and only two items which were unique to the scale (changing narrative constraints, denial/distortion of affect). A high score on the avoidant scale shows narratives in which children actively change the affective setting of the narrative by either denying it or by acting more aggressively. As the description of the scale is in line with attachment theory, other findings related to validity suggest that the avoidance scale also has its strengths. First, it relates negatively to the secure attachment scale in both groups, even though only one item overlapped between both scales in opposite directions (adults unaware). The item “adults unaware” provides empirical support for the theory that adults who can keep their children “in mind” provide more security than adults who are unaware of the problems [6,34,67,73,78]. Second, an interesting difference between settings was found relating to avoidance and distress/anxiety in the clinical group but not in the nonclinical group. For nonclinical children, this finding is in line with regular attachment theory, as less acknowledgment of distress or anxiety is characteristic for avoidant narratives. In contrast, children with higher ratings on the avoidance scale in the clinical group seemed to have more turmoil in this regard, just like the other children at intake. It might be that the avoidant children make more active use of neutralization of affect while entering the clinic, which is one of the items of the distress/anxiety scale. Third, three overlapping items have been found for the ambivalent and avoidant scales: “no closure,” “coherent aggression,” and “child shows aggression.” This suggests that these children are struggling with the attachment conflict and have difficulty solving it in a healthy manner. Instead, they get involved in aggressive conflicts that they try and solve using coherent aggression. In both insecure strategies, externalizing behavior is visible in the items which has been related to insecure attachment in other studies [58]. Interestingly, mothers in the nonclinical group rated their children with high avoidance as having conduct problems. In the clinical group, mothers of avoidant children did not report more child behavioral problems. For the three organized attachment classifications (secure, ambivalent, avoidant), our results support that the three organized attachment scales are not necessarily predictive of the presence or type of psychological and behavioral problems [18].

The empirical findings for the disorganized scale provide evidence for the reliability and validity of this theoretically constructed scale in the LPN-coding system. A high score on this scale shows the lack of a coherent strategy and responses that relate problematic and pathological states of mind with extreme or bizarre plots related to the conflict or role reversal with adults. Even though the disorganized attachment scale did not have any overlap of items with the other three scales, it showed interrelations in the expected direction, with negative associations with secure attachment and positive relations with the two insecure attachment scales across settings. Furthermore, it differentiated between the clinical and nonclinical group and was even the sole discriminatory attachment scale for girls in the expected direction. In addition, the distribution into attachment classification showed that 32% of the clinical group had a high disorganized score in contrast to 11% in the nonclinical group [75]. However, results for the assessment by the mothers of their children’s behavioral problems were more complicated. In the nonclinical group, children in the high-score group (median split) of disorganized attachment had mothers who reported more withdrawn behavior. In the clinical group, an interesting pattern occurred, with mothers reporting less problems for their children with higher disorganized attachment scores, in particular fewer internalizing problems, which is in contrast with previous findings [56,70]. It appears that these mothers have no mental image of their children’s inner world and only look at external behavior or normative rules of conduct. In addition, it may be the case that they are in denial of their children having internalizing behavioral problems. Often, these mothers will label emotional outbursts, such as crying, as a behavioral problem instead of an emotional display of distress. Furthermore, these mothers often have their own psychological problems and unresolved attachment issues that color their assessment of the children [31,79]. An assessment of the reflective functioning of parents, for instance with the Parental Reflective Functioning Questionnaire, might provide more context when discussing the results of the assessment [80].

Several methodological limitations need to be addressed, including potential directions for future research. First, coherence was rated on the same story stems and by the same rater that coded the LPN-items. It would have been methodologically stronger if these had been independently rated for each child, for instance, using other assessments. Nevertheless, adding coherence as part of the coding helped to assess the responses of children to each narrative and qualify their manner of dealing with the attachment conflicts from a more top-down perspective, without needing to be trained into assigning attachment classifications. For now, it seems advisable to include a coherence rating scale.

Second, although gender differences for attachment are not often found [81], it appears that using narratives may show a developmental difference between boys and girls in their verbal fluency or cognitive use of scripts as a defensive strategy in dealing with attachment issues [22]. This is in line with findings that continuous coding seems more sensitive to gender differences than categorical ratings [51]. Perhaps, this might solely be explained by developmental differences between boys and girls visible due to continuous scores. However, it might also relate to differences in how parents’ reminiscence about autobiographical memories with boys and girls emerges [15], which could have clinical implications. For clinical applicability, normative reference groups, based on continuous scores, could take these gender and age differences into account.

Third and relatedly, the use of continuous scales to examine the distribution of children into the four attachment categories was based on a ranking method, often used to produce normative data as a reference of what a “normal” range is in a particular population. Other methods could have been used. It is interesting to further explore the issue of creating complex profiles of continuous attachment scores. It might also be clinically relevant to have a method that is sufficiently sensitive to developmental and contextual change as alternatives for a categorical approach in middle childhood.

Fourth, the CBCL was only administered to mothers and the use of other informants, for example teachers, could have produced different results in relation to behavioral problems. For now, adding the CBCL did not produce very clear results in the clinical group. We suggest to include assessments about the current level of psychopathology of the parents (e.g., GHQ-12), or their level of reflective functioning, to investigate possible biases in their perception of their children’s problems.

Fifth, the three empirical attachment scales are based on results from the clinical population. It would be relevant to conduct a Confirmatory Factor Analysis on assessments in other clinical samples with the same age rage in middle childhood to investigate the robustness of the scales in larger diagnostically diverse populations. It would be interesting to combine results from different clinical practices who use and are trained in this method to investigate the reliability and validity of this measure in other clinical groups.

Finally, we did some exploratory analyses with the new distress/anxiety scale. The distress/anxiety scale appears to provide reliable and valid results [33], but more research is needed to discern if this is a relevant construct to assess either openness about their own emotions and emotions of others as characteristic of securely attached children or over-preoccupation with emotional issues and emotion dysregulation in relation to ambivalent attachment.

## 5. Conclusions

The present study has shown that a bottom-up scoring of attachment-relevant content items can produce three reliable and valid organized attachment scales. In addition, our findings support the theory that disorganized attachment is present in children alongside one of the organized attachment classifications as a separate dimension. The approach of assessing attachment continuously instead of categorically is clearly worth further investigation, considering the need for more psychometrically-sound attachment measures in middle childhood [18]. In addition, there is a definite need in clinical practice to understand the internal working model of children in middle childhood as an entry point for treatment and to point parents in the right direction concerning possible improvements in the quality of relating to and handling their children’s specific attachment issues. When mental health professionals are able to better predict and inform themselves and parents about the best way to approach and improve the relationship for children with insecure attachment histories, this may reduce communication errors, decrease mentalizing problems in relational interactions, and increase their emotional competence to deal with new learning experiences and challenges during adolescence.

## Figures and Tables

**Figure 1 ijerph-19-09053-f001:**
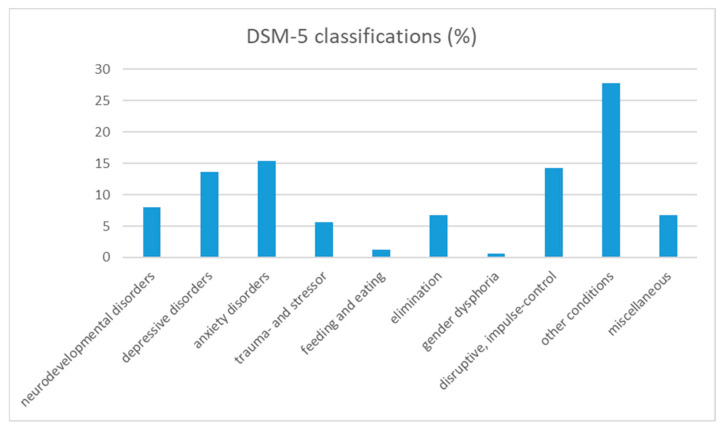
Distribution (%) of DSM-5 Classifications in the Clinical Group.

**Table 1 ijerph-19-09053-t001:** Means and Standard Deviations of the three main CBCL scales for the Clinical and Nonclinical Groups.

CBCL-T-Scores	Clinical *n* = 146		Nonclinical *n* = 96	
	*m*	*SD*	*m*	*SD*
Internalizing	64.3	9.54	49.1	10.04
Externalizing	61.3	10.54	45.9	8.52
Total score	64.9	8.82	47.3	9.11

*Note*. CBCL reports of mothers excluding missing data.

**Table 2 ijerph-19-09053-t002:** Results From a Factor Analysis of Items of the Little Piggy Narrative (LPN) Coding System.

		Factor Loading		
		1	2	3
Factor 1: Insecure-ambivalent				
4.	No closure *	**0.44**	−0.01	0.62
11.	Child endangered	**0.73**	0.01	0.26
12.	Child injured/dead	**0.73**	0.09	−0.20
19.	Adult actively rejects	**0.54**	0.15	−0.07
20.	Adult injured/dead	**0.81**	−0.09	0.08
22.	Physical punishment	**0.54**	−0.02	−0.11
23.	Child shows aggression *	**0.60**	−0.07	0.54
24.	Adult shows aggression	**0.79**	0.02	0.32
25.	Coherent aggression *	**0.62**	0.05	0.55
33.	Repetition	**0.43**	0.07	0.23
36.	Throwing away/out	**0.55**	−0.23	0.15
Factor 3: Insecure-avoidant				
2.	Disengagement (R)	−0.11	**−0.41**	0.15
3.	Initial aversion (R)	−0.06	**−0.48**	−0.08
5.	Premature foreclosure (R)	−0.24	**−0.70**	−0.04
8.	Child seeks help/protection	−0.03	**0.47**	−0.05
9.	Siblings/peers help	0.06	**0.46**	0.02
15.	Adult provides comfort	−0.05	**0.62**	0.02
16.	Adult provides help/protection	−0.20	**0.70**	−0.14
17.	Adult shows affection	0.14	**0.50**	−0.20
18.	Adult unaware (R) *	0.01	**−0.48**	0.45
21.	Limit setting	0.09	**0.43**	0.02
35.	Pleasurable domestic life	−0.21	**0.59**	−0.01
Factor 3: Insecure-avoidant				
4.	No closure *	0.44	−0.01	**0.62**
6.	Changing narrative constraints	−0.09	−0.30	**0.73**
18.	Adult unaware *	0.01	−0.48	**0.45**
23.	Child shows aggression *	0.60	−0.07	**0.54**
25.	Coherent aggression *	0.62	0.05	**0.55**
32.	Denial/distortion of affect	0.06	0.08	**0.67**

*Note. N* = 162. The extraction method was principal component analysis with a varimax (Kaiser normalization) rotation. Asterisk (*) for LPN-item that loads on two factors. Factor loadings above 0.40 are in bold. Reverse-scored items are denoted with (R).

**Table 3 ijerph-19-09053-t003:** Internal Consistency and Number of items of the Constructed Scales for the ASCT-LPN-coding system.

Constructed Scales	# of Items	Cronbach’s Alpha
Secure	11	0.76
Ambivalent	11	0.87
Avoidant	6	0.75
Disorganized	6	0.76
Distress/anxiety	3	0.74

**Table 4 ijerph-19-09053-t004:** Means, Standard Deviations, and *t*-test Statistics for the Scales comparing Younger and Older Children in Middle Childhood.

Variable		4–7 Years (*n* = 80–84)		8–10 Years (*n* = 76–78)		*t* (157–160)	*p*	Cohen’s *d*
		*M*	*SD*	*M*	*SD*			
Attachment scales								
	Secure	0.76	0.12	0.86	0.11	−5.48	<0.001	−0.87
	Ambivalent	0.20	0.17	0.19	0.18	0.66	0.51	0.10
	Avoidant	0.24	0.18	0.15	0.12	1.39	<0.001	0.53
	Disorganized	0.16	0.16	0.15	0.17	0.62	0.54	0.10
Other scales								
	Coherence	4.07	1.15	5.10	1.30	−5.27	<0.001	−0.85
	Distress/anxiety	0.37	0.27	0.50	0.32	−2.78	0.006	−0.44

*Note. N* = 156–162.

**Table 5 ijerph-19-09053-t005:** Means, Standard Deviations, and t-test Statistics for the Scales comparing Boys and Girls in the Clinical Group.

Variable		Boys (*n* = 96–99)		Girls (*n* = 60–63)		*t* (154–160)	*p*	Cohen’s *d*
		*M*	*SD*	*M*	*SD*			
Attachment scales								
	Secure	0.78	0.12	0.84	0.13	−3.12	<0.001	−0.51
	Ambivalent	0.22	0.19	0.16	0.14	2.33	0.021	0.38
	Avoidant	0.22	0.18	0.16	0.12	2.27	0.024	0.37
	Disorganized	0.18	0.19	0.12	0.10	2.40	0.018	0.39
Other scales								
	Coherence	4.31	1.32	4.99	1.23	−3.24	0.001	−0.53
	Distress/anxiety	0.36	0.29	0.54	0.31	−3.76	<0.001	−0.61

*Note. N* = 156–162.

**Table 6 ijerph-19-09053-t006:** Means, Standard Deviations, and *t*-test Statistics for the Scales comparing 8–10-year-old Boys and Girls in the Nonclinical Group.

Variable		Boys *(n* = 35)		Girls (*n* = 46)		*t* (79)	*p*	Cohen’s *d*
		*M*	*SD*	*M*	*SD*			
Attachment scales								
	Secure	0.87	0.09	0.92	0.09	−2.53	0.013	−0.57
	Ambivalent	0.17	0.17	0.13	0.12	1.24	0.217	0.28
	Avoidant	0.15	0.10	0.13	0.11	0.91	0.365	0.21
	Disorganized	0.07	0.16	0.02	0.04	1.76	0.082	0.40
Other scales								
	Coherence	5.37	0.99	5.80	0.79	−2.17	0.033	−0.49
	Distress/anxiety	0.54	0.59	0.66	0.56	−0.92	0.361	−0.21

*Note*. *N* = 81.

**Table 7 ijerph-19-09053-t007:** Correlations between the ASCT-LPN scales for Clinical and Nonclinical Groups.

	Secure		Ambivalent		Avoidant		Disorganized		Distress/Anxiety	
Secure			0.01		−0.35	**	−0.02		0.35	**
Ambivalent	−0.04				0.71	**	0.63	**	0.20	*
Avoidant	−0.50	**	0.56	**			0.46	**	0.04	
Disorganized	−0.13		0.72	**	0.45	**			0.13	
Distress/anxiety	0.30	**	0.43	**	0.20	*	0.30	**		

*Note.* * *p* < 0.05, ** *p* < 0.01. Below diagonal clinical group (*N* = 159–162), above diagonal nonclinical group (*N* = 98).

**Table 8 ijerph-19-09053-t008:** Correlations between the four Attachment Scales and Coherence.

Attachment Scales	Coherence			
	Clinical Group (*N* = 156–162)		Nonclinical Group (*N* = 98)	
Secure	0.61	**	0.52	**
Ambivalent	−0.48	**	−0.53	**
Avoidant	−0.53	**	−0.73	**
Disorganized	−0.61	**	−0.44	**

*Note.* ** *p* < 0.01.

## Data Availability

No supplementary or additional files included.

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
