# Peer review of "Attachment Stories in Middle Childhood: Reliability and Validity of Clinical and Nonclinical Children’s Narratives in a Structured Setting"

_ijerph, 2022, doi:10.3390/ijerph19159053_

Round 1

Reviewer 1 Report

Thank you for the opportunity to review this study entitled “Attachment Stories in Middle Childhood: Reliability and Validity of Clinical and Nonclinical Children’s Narratives in a Structured Setting” (ijerph-1750921).

The study aimed at exploring the psychometric properties of the a Dutch version of the Story-Stem Battery coded using the Little Piggy Narrative Coding System. A clinical (N = 162) and a nonclinical (N = 98) groups of children were involved in the research.

In my opinion, the research topic is relevant, and the study is interesting. Parallelly, there are some issues that need to be addressed before the paper will be suitable for publication.

1.     Please, reformat the bibliographic references (both the in-text one and those in the reference list) following the IJERPH guidelines.

2.     Page 6, lines 293, 294: “study participants comprised of two groups of children from the age of 4 years, 0  months to 10 years, 11 months.” Please rewrite this sentence to improve clarify. Furthermore, this seems inconsistent with the abstract, which refers to 4–10-year-old children.

3.     Please avoid references in the results section and use those bibliographical references in the discussion.

4.     The CFA analysis should be performed.

5.     Complementary to the limitations, directions for future research should be indicated.

6.     Please add a “Conclusions” section, which is indicated as mandatory in the IJERPH guidelines.

Reviewer 2 Report

The work submitted for review is an important contribution to the field of developmental psychology. The work shows high cognitive value, because attachment systems are a very important aspect of childhood, which often affects the entire further life of a person. As the authors point out, there is no special standard here, and psychometric tools designed to measure attachment should show high reliability.

In my opinion the paper is written correctly and its layout is appropriate. It reads well and all parts are arranged in a logical whole. Nevertheless, I have a few minor comments before the paper is accepted by the journal. I would suggest shortening the introduction, because in its current form it raises many issues that, in the case of original work, distort the reception of the work. At this point I would also suggest clearly signaling the purpose of the paper and the research hypotheses (they can be in the form of questions). In the methodology section, I suggest clearly stating the eligibility criteria for both groups - as a separate paragraph (labeled as part of the methodology). In Table 3 there is a different font, with the rest I suggest to unify the methods of presenting the results, because I have the impression that they have not been thought through enough. Please use the same design for tables in each case and take care of their layout and readability. It also concerns some inconsistency in the formatting of numerical data - sometimes there is a 0 before the dot, e.g. 0.37, and sometimes there is only .27 - please standardize it. I would drop Figure 1 and replace the information from it with a description. Also, the word "gender", please consider in what context you are using this word, if it is cultural then OK. If biologically default (dichotomous) I suggest the word "sex". In lines 781-785 something happened to the formatting of the manuscript. At the end of your paper, please include the strengths and weaknesses of the research you conducted and clearly highlight conclusions that may correspond with your hypotheses.

Thank you for allowing me to review this paper, and congratulations to the authors!

Reviewer 3 Report

Many thanks to the editor for giving me the opportunity to read this article. The study investigated the psychometric qualities of the attachment story stem battery coded using the Little Piggy Narrative Coding System in a clinical and a nonclinical group of children in middle childhood. It was enriching for me from a scientific and clinical point of view. As the authors rightly argue there is a definite need among clinicians to understand the internal working model of children as an entry point for treatment and to point parents in the right direction concerning possible improvements in the quality of relating to and handling their children’s specific attachment quality; to better predict and inform clinicians and parents about improving the relationship for children with insecure attachment histories, this may reduce communication errors, decrease mentalizing problems in relational interactions, and increase the emotional competence to deal with new learning experiences and challenges during adolescence. The topic was also covered in part in the following paper: doi: 10.1007/s40519-021-01282-6;  The present paper provided an empirical contribution by examining the usefulness of the attachment story stem narratives as an assessment instrument from early to late middle childhood (4-10-year-old children) for identifying the four attachment classifications: secure, ambivalent, avoidant and disorganized. The Introduction section is thorough and well structured, and the methodological and statistical section is very good; the assumptions are clear and accurate. The authors investigated: 1. the underlying constructs of the coding system in a clinical population by assigning the items to attachment scales that resulted from carrying out a Principal Component analysis and by relating these empirically derived scales to theory informed LPN-attachment constructs; 2. the internal consistency of the constructed scales and the interrater reliability of the LPN coding system; 3. the discriminatory and construct validity of the constructed scales examining differences between age groups, gender (males/females), and setting (clinical and nonclinical) for the constructed scales and coherence. I recommend putting the acronym in the abstract right after Little Piggy Narrative (LPN) - Coding System. I noticed that in Table 1 it is not made explicit whether the differences between the two groups are significant or not. If the tests were administered only to mothers I think this should be included in the limitations of the study. Overall the statistical analysis and results seem to me to be well described and the present research has shown that 1) a bottom-up scoring of attachment-relevant content items can produce three reliable and valid organized attachment scales; 2) the theory that disorganized attachment is present in children alongside one of the organized attachment classifications as a separate dimension. The approach of assessing attachment continuously instead of categorically is very interesting and useful. Using continuous variables, the issue of creating complex profiles in the child that go beyond a categorical approach could be explored. It would also be interesting - within future clinical and epidemiological research - to explore these issues from a large diagnostic and/or transdiagnostic group perspective.

Reviewer 4 Report

This is a preliminary study of the ASCT, looking at its factor structure, reliability, and validity. The authors have done an extensive literature review of the assessment instrument and how it has been used. They have a good sample size and two very distinct populations to assess with this instrument. Their justification for doing the project is that there are few assessment tools that determine attachment categories during middle childhood, and that those that exist (e.g., the ASCT) is not well-researched. Since this article was submitted to be in a special issue for IJERPH on assessment and diagnostic tools for developmental psychopathology, the authors have validated their measure using both a clinical and nonclinical population.

I have extensive comments on the attached document. Some of these comments are minor grammatical errors, some are clarity issues, and some are more conceptual issues. Overall, I would like to see a better justification for this study from a clinical perspective. Why is assessing attachment quality an important factor in clinical work with children? How does this attachment relate to problem behaviors as identified in the developmental psychopathology literature? Why use a projective (performance-based) assessment tool? Why not a set of questions the child can answer? The authors do a good job of identifying some of the qualities we look for in these stories that reflect attachment patterns. This needs to be tied into the qualities assessed in the ASCT. A lot can be cut out of the Intro so that other sections can be beefed up. Most importantly, there needs to be more clarity about the coding of the stories and the scales developed. I didn't realize until the Results section that the attachment scales were continuous. I thought the kids were categorized after the FA was done. Why was everyone given a continuous score of all the attachment categories? What is the hypothesis here? And, once the items were determined for each category, why weren't the categories used in more analyses?

One other comment about the Conclusions - you never mention how long/difficult it is to code these stories. How can you make this task more clinician-friendly? Are there certain key stories that were more robust in terms of categorizing them into the attachment categories? How might a clinician interpret some of these stories? Think about the gender findings as well - what are these findings saying about how pathology looks for boys and girls, and how parents might react to problems in boys and girls differently.

Author Response

Please see the attachment and also in this place, many thanks for your useful and thorough review!

Reviewer 5 Report

The manuscript reports an interesting paper on the reliability and validity of 2 tools for attachment evaluation in children. The goals of the manuscript are interesting, the methods are well described, however I have some suggestions for the authors that should be considered in order to improve their paper. 

- the introduction is too long. I think the authors should consider a revision due to the difficulty to follow all the information reported. A more focused introduction could be helpful in understanding the goals.

- why the ethical evaluation was required only for the non-clinical group? Is there any specific reason? 

- were non-clinical participants evaluated for psychiatric/psychological conditions?

- have you evaluated the distribution of the data? Because you used both parametric and non-parametric tests.

- is there any possible effects of the presence of more boys than girls in the clinical group?

- have you performed a prior power analysis to understand the group size needed? 

- is there any possible self-selection bias about the inclusion process? 

- I think that also the discussion should be revised and simplified, trying to be less speculative. 

- I think a native English speaker should evaluate the style of the writing that does not convince me. 

Reviewer 6 Report

 Abstract

The methodology is not described and no conclusions appear. Use of acronyms, not advisable in summary. The full name and acronym (LPN) do not appear together.

 Introduction

There is no clear review of the theoretical models underlying the research.

It does not explain the relevance of the sections, Non-clinical or typically developing participants (p.187) and At-risk or clinical participants (p.226).

Goals

A wide range of objectives and hypotheses (unnumbered and mixed) appear throughout the introduction. It is recommended that the objectives and hypotheses appear numbered.

Methodology

It is not described what type of methodology the research corresponds to. Nor is the research design clearly detailed: groups, conditions, variables analyzed, number of participants per group.

About description shows

It's confusing. The number of participants by groups (clinical/non-clinical) and ages are unclear. The categories used are neither precise nor justified (early childhood, early childhood, and late childhood). It is also quoted.

It would be more rigorous to use chronological ages. The months produce confusion, it is better to describe the age intervals by chronological years.

Instruments

The 671 Infant Attachment Interview (CAI) method is not cited or described.

Results

It is recommended that the results be ordered according to objectives and hypotheses

Discussion

The results are not discussed in an orderly manner; by objectives and hypotheses.

Neither the 671 Infant Attachment Interview (CAI) method nor the ABCD Model is described.

The statement appears: we built an extra scale called anguish/anxiety: Shouldn't it be a goal?

Use and inaccuracy of the terms early childhood and late middle childhood. Employment chronological ages.

Why can the contributions only be useful to doctors? And to psychologists and other professionals?

Conclusions.

There is no conclusion section.

Formal aspects

The rules of the review are not respected when citing and referencing the different studies reviewed.

The different works are not cited or referenced according to the regulations of the journal.

writing style

Non-precise terms: early childhood, early childhood, middle, late childhood. terminological confusion

Excess of initials and acronyms. Sometimes the acronyms are missing (eg in the first line of Discussion) and sometimes the acronyms are missing the full name.

Round 2

Reviewer 4 Report

The authors made significant improvements to this paper. They justified (and explained) the use of projective measures in a clearer way. The coding of these stories seems complex and categorizing into attachment groups even more complex. I think it's clearer how the continuous data was converted to categorical and why this was done. The authors' conclusions are also better justified. I have attached some minor edits.

Author Response

Dear reviewer,

Thank you for your kind words and constructive comments! They have been very helpful in improving the paper.

Please see the attachment for our responses to your minor edits.

Reviewer 5 Report

The authors have seriously improved the paper. They have addressed all my concerns. I think the manuscript can be accepted In the present form. 

Author Response

Thank you very much for your approval and of course for your very helpful comments.

Reviewer 6 Report

The authors have not corrected the main recommendations, referring to the objectives, results, discussion and conclusions.

The text sent has major legibility problems and understanding is very difficult. The strikethroughs and the different colors used by the authors (red, black and green) make it difficult to read and understand the text. The final text is not clearly discriminated. Therefore, the recommendation is to reject this article.

Author Response

Dear reviewer,

As far as we know, we did not submit a revised text with different colors or strikethroughs. Only in the file with a reply to your comments, we have used a yellow highlight to show the difference between your comment and our response. We did use your comments to improve the paper, for which we are grateful.